# Adverse Events in Home-Care Nursing Agencies and Related Factors: A Nationwide Survey in Japan

**DOI:** 10.3390/ijerph18052546

**Published:** 2021-03-04

**Authors:** Noriko Morioka, Masayo Kashiwagi

**Affiliations:** Graduate School of Health Care Sciences, Tokyo Medical and Dental University, Tokyo 113-8519, Japan; kashiwagi.fnls@tmd.ac.jp

**Keywords:** patient safety, adverse events, home healthcare, home-care setting, home-care nursing

## Abstract

Despite the importance of patient safety in home-care nursing provided by licensed nurses in patients’ homes, little is known about the nationwide incidence of adverse events in Japan. This article describes the incidence of adverse events among home-care nursing agencies in Japan and investigates the characteristics of agencies that were associated with adverse events. A cross-sectional nationwide self-administrative questionnaire survey was conducted in March 2020. The questionnaire included the number of adverse event occurrences in three months, the process of care for patient safety, and other agency characteristics. Of 9979 agencies, 580 questionnaires were returned and 400 were included in the analysis. The number of adverse events in each agency ranged from 0 to 47, and 26.5% of the agencies did not report any adverse event cases. The median occurrence of adverse events was three. In total, 1937 adverse events occurred over three months, of which pressure ulcers were the most frequent (80.5%). Adjusting for the number of patients in a month, the percentage of patients with care-need level 3 or higher was statistically significant. Adverse events occurring in home-care nursing agencies were rare and varied widely across agencies. The patients’ higher care-need levels affected the higher number of adverse events in home-care nursing agencies.

## 1. Introduction

Given policy changes that have shifted care from hospital to community settings in aging societies, the importance of patient safety at home has increased, since it enables people with diseases and/or disabilities to remain in their community [1,2]. Approximately 10.0–37.7% of patients using home-care services experienced adverse events (AEs) [3,4,5,6,7], which can lead to unnecessary or avoidable hospitalization [8]. To prevent AEs, home-care nurses play an important role as frontline healthcare providers who conduct the management and screening of safety risks [9]. Inadequate knowledge and assessment skills, however, increase the risk of AEs [10]. A lack of continuing education and failure to update routines and procedures have been found to compromise organizational patient safety culture [11]. Despite their importance, the evidence of AEs in home-care settings, especially related to home-care nursing agencies, is still limited [1,10,11].

In Japan, home-care nursing services are covered by the National Health Insurance System or the long-term care insurance (LTCI) system [12]. In this service, home-care nurses visit patients’ homes and provide the following nursing care: assistance for activities of daily living (ADLs) (e.g., bathing and grooming), rehabilitative training, wound care (e.g., bedsores), end-of-life care, management of catheters and other medical devices, among other services. In the 2019 fiscal year, approximately 5.8 million patients received home-care nursing services, and this figure has been increasing annually [13]. The majority of individuals who receive home-care nursing services are adults aged 65 years or older requiring long-term care due to chronic disease or disability [12,13]. Only the agencies designated by the prefectural governor or the mayor of a designated municipality can deliver home-care nursing services covered by public insurance. To be designated agencies, the agencies should meet equipment, staffing, and operational standards (i.e., with 2.5 or more full-time equivalent nurses) and have an administrator with a nursing or public health nursing license. As a juridical person, both for-profit companies and public organizations can open these home-care nursing agencies [12].

A case report in 2004, with 157 incidents occurring during home-care nursing service delivery, showed that 15% of incidents lead to hospitalization and death [14]. Consequently, home-care nursing agencies have engaged with patient safety. Almost all home-care nurse agencies have their own reporting system regarding AEs within the agency [14]. Although long-term care acts mandated agencies to report incidents to the local government [15], the format is not standardized but rather personalized by each local government, and no nationwide standardized reporting system regarding AE is available in Japan.

Despite the recognition of the importance of patient safety in home-care nursing, little is known about the nationwide situation of incidence of AEs and associated factors among home-care nursing agencies in Japan. There is only one survey conducted in 2004 available to describe the nationwide incidence of AE in a home-care nursing agency [16]. According to the survey results, the mean number of AEs in one month was 0.15, and almost 70% of the agencies experienced no AEs. However, the factors related to the occurrence of AEs were not analyzed. In Japan, the number of home-care nursing agencies has drastically increased from 5000 agencies in 2004 to approximately double that in 2020 [17]. Patient safety management and the occurrence of AEs in home-care nursing agencies has been changing. A better understanding of the current nature of AEs and the underlying causes of these AEs in home-care nursing agencies will help to ensure delivery safety nursing care at home. Therefore, this paper describes the nationwide incidence of AEs among home-care nursing agencies in Japan and investigates the agency factors associated with AEs.

## 2. Materials and Methods

### 2.1. Design and Data Collection

We conducted a cross-sectional nationwide questionnaire survey. Self-administrative and anonymous questionnaires were mailed to all available home-care nursing agencies (9979 in total), the postal addresses of which were obtained from the Information Publication System for Long-Term Care database [18], at the end of March 2020. In Japan, all long-term care service agencies, including home-care nursing agencies, are required to report their status to local (prefecture) governments annually in accordance with the Long-Term Care Insurance Act (Article 115-35-44) [15], and the government discloses the list as the database via the website [18]. As of April 2019, 11161 agencies were functional [19]. The database we used in this study covered approximately 90% of the agencies. We asked the nursing administrators of the agencies to answer the questionnaire and return it by April 2020.

### 2.2. Measurements

We used an original questionnaire that included the number of AEs, the process of care for patient safety, and other agency characteristics. Three researchers in home-care nursing and nursing administration developed the initial items by conducting a literature review and a focus group interview composed of four nursing administrators of home-care nursing agencies. The content validity of the questionnaire was assessed by an expert panel which included patient safety management experts, a home-care doctor, home-care nurses, researchers in home-care nursing, and a policymaker.

#### 2.2.1. Number of Adverse Events

The World Health Organization (WHO) defines an AE as “a harmful incident or an incident that results in harm to a patient” [20]. However, the definition of AE in home-care settings varies across studies [2,3,4,7,8,21]. In this study, we used an original instrumental definition of AE in accordance with the WHO definition, previous studies, and experts’ opinions in accordance with the actual situation of Japanese home-care nursing services. An AE was defined as an event that results in unintended harm to the patient, occurring during service delivery, including any level of severity. AEs consisted of falling down, aspiration, medication error, device-related error, tube-related error, pressure ulcers, catheter related infections, and urinary tract infections (UTI). In the questionnaire, we asked the number of each type of AE that was detected during a three-month period in each agency (January 2020–March 2020). As mentioned above, there is no standardized reporting system regarding AEs. The nursing administrators of the agencies answered AE numbers and type every month (January 2020, February 2020, and March 2020) based on their own reporting system.

#### 2.2.2. Process of Care for Patient Safety

The process of patient safety care at the agency level was assessed by having a manual for patient safety, having a committee for patient safety, and providing training to nursing staff regarding the prevention of AEs. All questions were binary variables (yes or no).

#### 2.2.3. Agency Characteristics

The agency characteristics addressed in the questionnaire are acceptance of pediatric patients (yes or no); acceptance of patients at the terminal care stage (yes or no); the percentage of patients who need special medical treatment; and the percentage of patients with care-need level ≥3, type of agency ownership, number of patients in a month, number of full-time equivalent nurses, among other factors. Patients who need special medical treatment included those with cancer, pain control, oxygen therapy, injection and IV, central venous hyperalimentation, nasal tube feeding, enema and stool extraction, or pressure ulcers beyond the dermis. In Japan, the care-need level is classified into six levels: support required and care required (care levels 1–5). Care-need level 3 or over means moderate to higher care-need level, and those with car-need level 3 or over are allowed to use institutional care services (i.e., nursing homes) in the LTCI [22].

### 2.3. Statistical Analysis

We conducted a complete case analysis for missing data. We described the summary of the incidence of AEs and agency characteristics. To investigate the generalization of our sample, we compared our sample’s characteristics to national statistics from the Survey of Institutions and Establishments for Long-term Care in 2018 [23]. In the national statistics, we only obtained data for region, types of agency ownership, and the number of full-time equivalent nurses.

To investigate the factors related to the incidence of AE, we used zero-inflated negative binomial (ZINB) models for count data. First, the Poisson model was rejected because the distribution of the number of AEs was over dispersed, which means that the deviation was over the mean. In addition, as the zero cases of AEs were inflated in our sample, a negative binomial model did not fit our data. As the negative binomial model attempts to account for the high number of zeros and the counts simultaneously, the predicted values were overly biased toward the zeros and the residual variation was high [24]. Therefore, we used the ZINB model. The ZINB model consists of two parts: a negative binomial part predicting the number for those agencies that are not “certain zeros”, and a logit part that is generated for the “certain zero” cases, predicting whether an agency would be in this group. In the ZINB model, the expected number of AEs changes by exp. (coefficient) for each unit increase in the corresponding predictor [24]. We selected variables in the models from the literature review. The Akaike information criterion (AIC) suggested the use of ZINB rather than the Poisson regression model, the negative binomial regression model, and the zero-inflated Poisson regression model. A two-tailed *p*-value  < 0.05 was considered statistically significant. All analyses were performed using Stata version 16 MP (StataCorp. College Station, TX, USA).

### 2.4. Ethical Considerations

This study was conducted in accordance with the Declaration of Helsinki (http://www.med.or.jp/dl-med/wma/helsinki2013e.pdf (accessed on 3 March 2021)), and the protocol was approved by the Medical Research Ethics Committee of Tokyo Medical and Dental University (No. M2019-304). Participants signed an informed consent form in the questionnaire to participate in the study.

## 3. Results

Out of 9979 agencies, 580 questionnaires were returned (participation rate 5.8%), and 180 were excluded due to incompleteness because of missing data, and 400 agencies were included in the final analysis. Table 1 shows the characteristics of the 400 agencies. The most common agency ownership type was for-profit (177 agencies). The median number of full-time equivalent nurses was 4.0. Among the 400 agencies, approximately 90% had a manual for patient safety, while 26.5% and 52.3% of agencies had a committee for AE prevention and training for AE prevention, respectively. Our study participants had almost the same characteristics as those in the national statistics (Appendix A, Table A1).

The number of AEs in each agency ranged from 0 to 47, and 106 (26.5%) agencies did not report any AEs among the 400 agencies (Figure 1). The median (25–75 percentile) number of AEs was three (0–7). In total, 1937 AEs had occurred over three months (Table 2). By types of AEs, the number of pressure ulcers was 1725 (80.5%), the number of UTIs was 127 (5.9%), and the number of falls was 36 (1.7%), in descending order (Table 2).

Table 3 shows the results of the univariate and multivariate ZINB models for AEs. In the inflated part in the univariate ZINB analysis, a higher percentage of users with care-need level 3 or more (the third group coefficient −1.70, *p* < 0.05), a higher percentage of users who needed medical treatment (the third group coefficient −1.17, *p* < 0.05; the fourth group coefficient −1.50, *p* < 0.01), and a higher number of patients in a month (coefficient −0.70, *p* < 0.01) were less likely to report zero AEs. In the negative binomial part of the univariate ZINB analysis, acceptance of pediatric patients (coefficient 0.32, *p* < 0.05), acceptance of terminal-care stages (coefficient 0.42, *p* < 0.01), a higher percentage of patients with care-need level 3 or more (the fourth group coefficient 0.35, *p* < 0.05), not having a committee for AE prevention (−0.35, *p* < 0.01), training for AE prevention (0.30, *p* < 0.05), and a higher number of patients in a month (0.35, *p* < 0.001) were statistically significantly associated with the occurrence of AEs.

After adjusting for the number of patients in a month as an exposure and other variables in multivariate ZINB analysis, the percentage of patients with care-need level 3 or over was statistically significant in the negative binomial part. The second quartile group had a number of AEs of exp. (0.36) ≒ 1.44 times (*p* < 0.05), the third quartile group had exp. (0.27) ≒ 1.32 times (*p* = 0.099), the fourth quartile group had exp. (0.37) ≒ 1.42 times (*p* < 0.05) higher than that of the agency in the first quartile. Having a committee for AE prevention gave exp. (−0.23) ≒ 0.71 times (*p* = 0.065) lower number of AEs than that of agencies without a committee for AE prevention, but this was not statistically significant.

## 4. Discussion

This study described the current nationwide situation of the incidence of AEs, including various types, at the home-care nursing agency level across Japan. We also found that AEs were more likely to occur in the agencies with more patients with higher care-need levels after adjusting for potential confounders.

In this study, AEs in home-care settings in Japan were found to be quite rare, with almost one quarter of the agencies detecting zero incidents. This low number of AEs is consistent with previous studies. In a prospective study with 419 users of home-care nursing services in Japan, the incidence and period prevalence rates of infectious diseases were 0.63% and 15.0%, respectively [25]. Another survey of home-care nurses reported that approximately 70% of home-care nurses did not experience AEs [26]. AEs related to home-care nursing seem to be rare in Japan. Hence, this result might be affected by agencies with poorly organized surveillance systems. Although almost all agencies have their own reporting system [14], 20% of those agencies do not save the documentation regarding the AEs [16]. In addition, the number of AEs varied widely. The wide variation in the occurrence of AEs might be explained by the fact that there is no standardized reporting system across Japan. One third of the agencies do not use a reactive approach to detect and prevent AEs [27], which may result in the underreporting of AEs. A scoping review showed that the overall reported AE rate in other counties was wide-ranging, and highlighted the necessity of a standardized process of data collections and reporting of specific AEs [3]. A nationwide standardized detection and reporting system is necessary for Japan as well.

Relating to the type of AE, pressure ulcers represented the majority of AEs in the current study. Generally, pressure ulcers are frequently detected in the home-care setting [1]. A previous study in Sweden found that the incidence of healthcare-associated infections was 21.8%, that falls was 18.8%, and that of pressure ulcers was 17.0% among all AEs [7]. In a Canadian nationwide study, the most frequent AEs were injurious falls (n = 16, 17.2%) and wound infections including pressure ulcers (n = 13, 14%) [5]. Compared to findings in other countries, this study indicated that the percentage of pressure ulcers has been increasing in Japan. This is likely explained by the patient characteristics and the home-care nursing system in Japan. First, more than 30% of home-care nursing service users in Japan are bedridden older adults with higher care levels [13], and immobility is a well-known risk factor for pressure ulcers [28]. Second, the fee-schedule in LTCI mandates that all agencies report the incidence of pressure ulcers to the local Bureau of Health and Welfare annually [29]. False reporting, if discovered, results in the agency having to refund the payment, which should prevent any deliberate falsification. All agencies, therefore, detected the number of pressure ulcers accurately, unlike other types of AEs.

Regarding the factors associated with AE, we found that the agencies with more patients with higher care-need levels tended to encounter more AEs. As previous studies have pointed out that low ADLs are a risk factor for AE [5], dependency on ADLs in the agency may affect the AE incidence. In our study, the agencies with a prevention committee were less likely to have AE, but the association was not statistically significant. Training for AE prevention was not associated with AE. Previous studies have shown that almost half of all AEs are preventable [5], and the process of nursing care, including continuing education and standardized care procedures, is important to prevent AEs [2,9,10,30]. Further study is necessary to investigate whether actual prevention practices in home-care nursing agencies contribute to patient safety.

Our study has several limitations. First, the response rate was extremely low (5.8%). This might be influenced by the COVID-19 outbreak in Japan. In Japan, COVID-19 spread in March 2020, which is almost the same time as our survey. We could not mail the reminder later because most administrators in agencies were struggling to deal with the surging number of COVID-19 cases across the nation during the study period. Despite this quite low response rate, the response rates by region were almost the same as those of national figures (Appendix A, Table A1).

Second, we asked about the incidence of AEs during the last three months using a self-reported questionnaire. Best practice agencies that collected an exact number of AEs and conducted well organized prevention practice might be likely to respond to our study. Our results might have been underestimated because of recall bias and social disability bias. Further research is necessary to investigate the use of chart reviews or standardized mandated reporting systems.

## 5. Conclusions

Our nationwide survey showed that the number of AEs occurring in three months in home-care nursing agencies was low and varied widely across agencies, with one quarter of the agencies not reporting any AEs in Japan. Pressure ulcers were the most frequently reported. After adjusting for confounders, AEs were more likely to occur in the agencies that delivered services for more patients with higher care-need levels.

## Figures and Tables

**Figure 1 ijerph-18-02546-f001:**
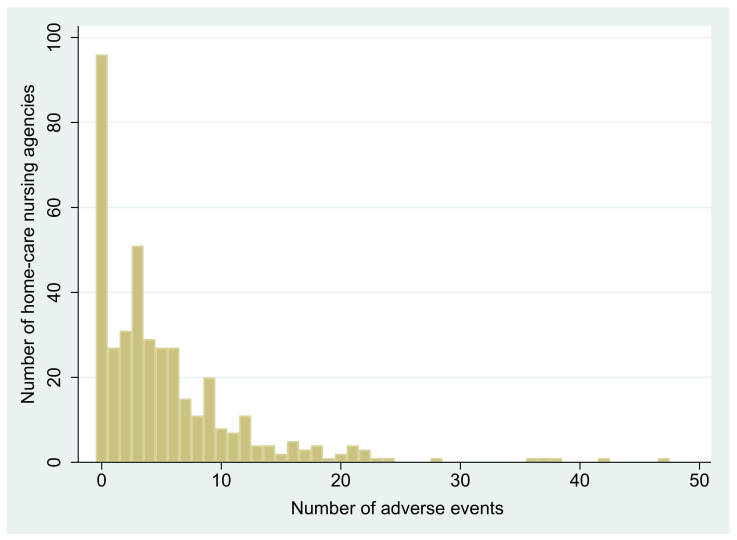
Distribution of number of adverse events among 400 agencies.

**Table 1 ijerph-18-02546-t001:** Characteristics of home-care nursing agencies (n = 400).

Variables		
Operating and management		
Agency ownership (n,%)		
Healthcare corporation	111	27.8
Profit	177	44.3
Social welfare	88	22.0
Others	24	6.0
Number of nurses (full-time equivalent) (median, 25—75 percentile)	4.0	3.0–5.8
Number of users in a month (median, 25—75 percentile)	53.5	29.5–79.5
Patient characteristics		
Accepts pediatric patient (n, %)		
Yes	92	23.0
No	308	77.0
Accepts patient at terminal care stage (n, %)		
Yes	105	26.3
No	295	73.8
Percentage of users with care-need level ≥3 (median, 25—75 percentile)	28.0	18.7–38.0
Percentage of users who needs medical treatment (median, 25—75 percentile)	16.3	6.8–27.5
Process of care for patient safety		
Having a manual for patient safety (n,%)		
Yes	357	89.3
No	43	10.8
Having a committee for adverse event prevention (n,%)		
Yes	106	26.5
No	294	73.5
Training for adverse event prevention (n,%)		
Yes	209	52.3
No	191	47.8

**Table 2 ijerph-18-02546-t002:** Number of adverse events by categories/types that occurred over three months.

	n	%
Total number of adverse events	1937	100.0
By types		
Pressure ulcers	1725	80.5
Urinary tract infection	127	5.9
Falling	36	1.7
Tube-related error	17	0.8
Catheter related infection	12	0.6
Aspiration	8	0.4
Medication error	6	0.3
Device-related error	6	0.3

**Table 3 ijerph-18-02546-t003:** Results of zero-inflated negative binomial regression models for adverse events.

	Univariate ZINB								Multivariate ZINB							
	Negative Binomial Part				Inflated Part				Negative Binomial Part				Inflated Part			
	Coef.	95% CI		*p*-Value	Coef.	95% CI		*p*-Value	Coef.	95% CI		*p*-Value	Coef.	95% CI		*p*-Value
Agency ownership (ref. healthcare corporation)																
Profit	0.01	−0.28	0.30	0.936	0.35	−0.44	1.13	0.386	−0.06	−0.34	0.23	0.700	0.34	−0.49	1.17	0.417
Social welfare	0.03	−0.29	0.35	0.859	−2.46	−7.66	2.74	0.354	−0.17	−0.50	0.16	0.319	−1.86	−6.05	2.32	0.383
Others	0.31	−0.19	0.81	0.222	−0.44	−2.18	1.30	0.621	0.18	−0.29	0.65	0.444	−0.33	−1.98	1.32	0.695
Accepts pediatric patient (ref. does not accept)	0.32	0.07	0.58	<0.05	−1.08	−2.33	0.16	0.089	0.10	−0.22	0.42	0.530	−0.04	−1.42	1.33	0.949
Accepts patient at terminal care stage (ref. does not accept)	0.42	0.17	0.66	<0.01	−0.62	−1.50	0.27	0.172	0.14	−0.12	0.40	0.297	−0.35	−1.40	0.70	0.515
Percentage of users with care-need level ≥3 (ref. first quartile)																
second quartile	0.33	−0.01	0.68	0.059	−0.76	−1.66	0.14	0.097	0.36	0.04	0.69	<0.05	−0.44	−1.48	0.61	0.412
third quartile	0.28	−0.06	0.62	0.109	−1.70	−3.18	−0.21	<0.05	0.27	−0.05	0.60	0.099	−0.90	−2.05	0.26	0.128
fourth quartile	0.35	0.01	0.70	<0.05	−0.91	−1.86	0.03	0.058	0.37	0.04	0.70	<0.05	−0.76	−1.77	0.25	0.142
Percentage of users who needs medical treatment (ref. first quartile)																
second quartile	−0.15	−0.50	0.19	0.389	−0.95	−1.94	0.03	0.059	−0.17	−0.50	0.17	0.334	−0.37	−1.39	0.66	0.485
third quartile	0.18	−0.15	0.52	0.281	−1.17	−2.16	−0.18	<0.05	0.08	−0.25	0.41	0.624	−0.52	−1.55	0.51	0.326
fourth quartile	0.30	−0.03	0.63	0.079	−1.50	−2.63	−0.36	<0.01	0.19	−0.15	0.52	0.270	−1.03	−2.19	0.14	0.085
Having a manual for patient safety (ref. without)	0.02	−0.34	0.38	0.917	0.70	−0.98	2.39	0.412	−0.07	−0.42	0.29	0.702	0.93	−0.72	2.59	0.270
Having a committee for adverse event prevention (ref. without)	−0.35	−0.60	−0.09	<0.01	−0.15	−0.89	0.59	0.696	−0.23	−0.47	0.01	0.065	−0.33	−1.13	0.47	0.418
Training for adverse event prevention (ref. no)	0.30	0.07	0.53	<0.05	0.47	−0.29	1.24	0.226	0.12	−0.11	0.35	0.307	0.64	−0.21	1.49	0.143
Log (number of patients in a month )	0.35	0.20	0.49	<0.001	−0.70	−1.17	−0.24	<0.01	0.26	0.09	0.44	<0.001	−0.79	−1.48	−0.10	<0.05

## Data Availability

The data are not publicly available due to ethical issue.

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
