# Peer review of "Adverse Events in Home-Care Nursing Agencies and Related Factors: A Nationwide Survey in Japan"

_ijerph, 2021, doi:10.3390/ijerph18052546_

Round 1

Reviewer 1 Report

It is evaluated that authors sufficiently revised the manuscript in accordance with review's opinion.

No need to further revisions.

Author Response

We sincerely appreciate your careful review and comments.

Reviewer 2 Report

Adverse events in home-care nursing agencies and related factors: A nationwide survey in Japan

Thank you for re-submitting your work and fully engaging with the review process, your paper has developed significantly. I have no further comments to add.

Author Response

(The authors gave the same response as above.)

Reviewer 3 Report

Dear Authors, I would like to suggest some changes to the manuscript. First of all, the abstract is clear and describes the content very well. As for the introduction, you present what is already known very clearly and then you outline the research question/ the aim of the study. The process of the sample selection and the methods, in general, should be described in detail and in a more precise way. Also, the discussion part had better be enriched with results of previous findings in this field and the authors in this way can make more sound comparisons. Additionally, another point that should be highlighted is the content of the questionnaire. In the text, you do not describe the tool you have used and the way that this questionnaire has been developed. According to the text, it is a new tool developed by the authors it should be mentioned and the whole process should be described in detail.

Author Response

Dear Authors, I would like to suggest some changes to the manuscript. First of all, the abstract is clear and describes the content very well. As for the introduction, you present what is already known very clearly and then you outline the research question/ the aim of the study.

>Authors’ response

We sincerely appreciate your careful review and comments.

The process of the sample selection and the methods, in general, should be described in detail and in a more precise way.

>Authors’ response

Thank you for the comment. This survey was not sample survey. We mailed questionnaire to all available home-care nursing agencies obtained from the national database. For missing data, we conducted complete case analysis. Out of 580 which were returned, 180 with missing were excluded. We, therefore, added the explanation as follows.

L71

Self-administrative and anonymous questionnaires were mailed to all available 9979 home-care nursing agencies

L116

We conducted a complete case analysis for missing data.

L142

Out of 9979 agencies, 580 were returned (participation rate 5.8%), 180 were excluded due to incompleteness because of missing data, and 400 agencies were included in the final analysis.

Also, the discussion part had better be enriched with results of previous findings in this field and the authors in this way can make more sound comparisons.

>Authors’ response

Thank you for the commend. As you pointed out, we added the comparison with the findings in other countries regarding current issue and challenge in patient safety management in home-care setting in the discussion part. Unfortunately, there is very limited evidence on AE in home-care setting. We already mentioned about available previous studies in the manuscript.

L196

A coping review showed that the overall reported AE rate in other counties was wide-ranged, and highlighted the necessity of a standardized process of data collections and reporting of specific AEs’ [3]. A nationwide standardized detection and reporting system is necessary for Japan as well.

Additionally, another point that should be highlighted is the content of the questionnaire. In the text, you do not describe the tool you have used and the way that this questionnaire has been developed. According to the text, it is a new tool developed by the authors it should be mentioned and the whole process should be described in detail.

>Authors’ response

Thank you for the comment. We developed an original questionnaire by conducting a literature review and a focus group interview comprised of four nursing administrators of home-care nursing agencies. The content validity was assessed by an expert panel. We, therefore, added the more detail explanation as follows.

L80

We used an original questionnaire that included the number of AEs, the process of care for patient safety, and other agency characteristics. Three researchers in home-care nursing and nursing administration developed the initial items by conducting a literature review and a focus group interview comprised of four nursing administrators of home-care nursing agencies. The content validity of the questionnaire was assessed by an expert panel which included patient safety management experts, a home-care doctor, home-care nurses, researchers in home-care nursing, and a policymaker.

Round 2

Reviewer 3 Report

Thank you for incorporating the comments and for making the suggested changes as well. The manuscript will be very interesting in the present form for the readers.

This manuscript is a resubmission of an earlier submission. The following is a list of the peer review reports and author responses from that submission.

Round 1

Reviewer 1 Report

Thank you for taking the time to review the paper.

The adverse event in home-visit nursing agencies in this study is critical issue. It will be easier to understand if a explanation about the home-visiting nursing care system in Japan is added in introduction. Especially, the information such as which target is for this service and how often the nurses visit to patients' home seems to be essential.

The data of 400 facilities were analyzed in this study out of about 10,000 facilities, the response rate is about 5%. So you should check the rate in discussion part (line 192).

If you can find the information about the total home-visiting nursing agencies in Japan, it would be nice to add a discussion on whether the 400 facilities in this study are similar to the characteristic of the entire home -visiting nursing agencies.

Reviewer 2 Report

Adverse events in home-visit nursing agencies and related factors: A nationwide survey in Japan

Thank you for submitting your work to International Journal of Environmental Research and Public Health and providing me with the opportunity to review your paper. I hope you find my comments both helpful and constructive as this is my aim.

Unfortunately there are grammatical and sentence construction errors that impact on the readability of your paper.

Title – consider revising the title, as it is unclear what you mean by ‘home-visit nursing agencies’

Abstract

  • Home-visit nursing is not a phrase we use in the UK and it is grammatically poor, would it be possible to re-consider this phrase
  • Clarity is required regarding ‘home-visit nursing agencies’ as it is unclear what you mean

Introduction

  • Your introduction needs to be developed further, for example you do not define AEs or a home visit, the population receiving home visits – the size of this population, perhaps the increase in this population leading to a higher demand

Data collection

  • How did you follow-up data collection, especially as your response rate was so low? More information is required here

Measurements

  • Were other measurements collected, but not included in this paper?
  • Could each institution only provide the number of each of the AEs you describe, without providing their own list?

Patient safety

  • Do you think your approach to identifying patient safety mechanisms is comprehensive? As not appear to include policies, process or guidelines, and if nurses completing home visits are registered nurses, surely there is a professional obligation that needs to be considered as well?

Agency characteristics

  • For readers outside of the Japan, how are these agencies funded?

Results

  • How generalisable are you results due to your poor response?

How can you assume consistency of reporting across agencies, especially as their responses was so diverse?

Reviewer 3 Report

Thanks for inviting me to review this research work about adverse events in home-visit nursing agencies and related factors. The following comments could be useful in the development of this document.

Abstract: Abbreviations used in the abstract are not allowed. Please, consider replacing it.

Introduction: Some of the statements made are not relevant to an international audience and should be reviewed, for example, comments about regulatory boards. To introduce the international perspective will helping. For example, what is the operational definition of "home-visit nursing agencies"? How do they work in your country? And in others? In some countries, maybe they do not exist this kind of healthcare services. Also, the development of rationale will increase the interest of the audience.

Material and Methods:

Lines 64-65: The Nursing administrators how completed the questionnaire. Did they look for the information in any kind of a reported event adverse database? Adverse Events Reported in any database? National? Regional? Local? Institutional? Did they ask to nurse staff or others? Clear explanations about this matter are critical. Please, add in detail.

Lines 75-77: Why did you select these AEs and refuse others? Where did you find this list of AEs that you have used?

Lines 83-93: Based on what evidence, previous studies, or criteria have these characteristics been selected and defined? Variables justification are missing.

Lines 95-105: It would be desirable to clarify this point. It is difficult to understand why researchers have selected zero-inflated negative binomial models (ZINB).

Line 107: Declaration of Helsinki cite is missing.

Results:

Lines 112-113: The participation rate is missing.

Lines 113-116: It would be advisable not to repeat in the text the results already included in the tables. Why does the text refer to these data in the table and not others? What is the rationale?

Lines 116-117: Match the numerical data to the concept to increase clarity.

Table 1: It seems the "n" or the "median" data are missing. Maybe "percentile" word is more common than "tile". Think about changing it. IQR? Use a footer to explain it could be a solution. Please check and complete this table.

Figure 1: The graph needs to include numerical data relative to percentages.

Table 2: Do not use the comma to indicate thousands. Follow the guide for authors. Presenting the types of adverse events according to more population variables, such as age and gender, would be interesting. The table results are a bit insufficient.

Line 138: It would be advisable to increase the clarity of the explanation in the text of the results shown in Table 3.

Discussion

Check and updated after previously suggested changes have been made.

Lines 194-196: Maybe be to include data about agencies participation by region in the results section will be needed in order to support this explanation.

Conclusion: Check and updated after previously suggested changes have been made. The conclusion should follow from the analysis and arguments presented.

References: Please, adapt them following authors guidelines for this journal.